# Study the Parametric Effect of Pulling Pattern on Cherry Tomato Harvesting Using RSM-BBD Techniques

Huaibei Xie [1,2,3], Deyi Kong [1,2,4,*], Jianhua Shan [5] and Feng Xu [1,2]

1   Institute of Intelligent Machines, Hefei Institutes of Physical Science, Chinese Academy of Sciences, Hefei 230031, China; hbxie@mail.ustc.edu.cn (H.X.); xf1992@mail.ustc.edu.cn (F.X.)
2   Science Island Branch of Graduate School, University of Science and Technology of China, Hefei 230026, China
3   School of Mechanical Engineering, Anhui University of Science and Technology, Huainan 232001, China
4   Innovation Academy for Seed Design, Chinese Academy of Sciences, Sanya 572024, China
5   School of Mechanical Engineering, Anhui University of Technology, Ma'anshan 243032, China; 2931@ahut.edu.cn
*   Correspondence: kongdy@iim.ac.cn; Tel.: +86-551-65593242

**Abstract:** Detachment of fruit from the plants with separation force is important in robotic harvesting. Compared with twisting pattern and bending pattern, the pulling pattern for cherry tomato harvesting is more simple, more flexible, and easier to implement in robotic harvesting. It was found that the detachment force is closely related to the location of the fruit separation. However, in the pulling pattern, analysis of the effect of harvesting parameters of cherry tomatoes at the calyx/fruit joint has still not been carried out in depth. In this paper, the goal of this research was to investigate the effect of different harvesting parameters on the minimal detachment force of cherry tomatoes at the calyx/fruit joint. Experiments were designed according to response surface methodology Box–Behnken design by maintaining three levels of three process parameters—grasping angle, horizontal angle, and pitching angle. Results showed that the pitching angle is the most important parameter, and the grasping angle has little effect on the detachment force, and the detachment force was found within the range of 0.58 N to 2.46 N. Results also revealed that the minimum separation force of the cherry tomato harvesting at the calyx/fruit joint was obtained by the optimum conditions of the grasping angle of 68°, the horizontal angle of 135° and the pitching angle of 0°. Moreover, desirability function has also been used to optimize the angle parameters. The confirmation experiments validate the reliability and capability of the developed model.

**Keywords:** plucking pattern; response surface; detachment force; cherry tomato; harvesting



## 1. Introduction

Global cherry tomato production exhibits a trend of continuous growth [1]. Fresh cherry tomatoes are in high demand because they are an important part of the diets of millions of people around the world [2]. Like many other fresh fruits, cherry tomatoes usually have to be harvested one by one, which is both a time-intensive and labor-intensive task that utilizes about 40% of the total cost until now, with workers requiring experience and skill [3]. Faced with the population growing negatively and the serious labor shortage in China, it is urgent to develop cherry tomato harvesting robots. Research on cherry tomato harvesting robots has been conducted since the 1990s [4,5]. However, due to high manufacturing costs, a low detachment efficiency, and environmental complexities, there are no commercial robotic systems for cherry tomato picking in the market [6,7].

For robotic harvesting, fruit detachment depends on branch–stem characteristics, phenotypic characteristics, and detachment techniques, as well as plucking patterns [8,9]. Detachment techniques are generally a combination of tension force, bending stress, and shear stress [10,11]. The pulling pattern is one of the most important plucking patterns:

it is used to grip the target fruit with the end-effector, then successfully detaches it from the plant [12]. The twisting pattern has the advantages of smaller gripping force, less fruit damage and lower fruit detachment success rate as well as lower harvesting efficiency. When the end-effector gripping force is low, the fruit may slip out of the end-effector during harvesting, which increases the risk of a low harvest success rate. However, the fruit detachment success rate and harvesting efficiency can be improved by optimizing end-effector and experimental schemes. Yaguchi et al. designed an end-effector for harvesting tomatoes with an approximate success rate of 60% by twisting pattern in approximately 23 s/fruit [13]. The bending pattern brings a risk of potential damage, which may bruise or even knock off adjacent fruits [14]. In the bending pattern, some woody peduncles may be hard to be bent off [5].

Compared with other picking patterns, pulling for picking cherry tomatoes is much easier to implement: the reason is that once the fruit is caught, it must be picked with a high success rate of harvesting [13]. However, the gripping force should be controlled reasonably when the target fruit is harvested in a pulling pattern, otherwise the fruit may be damaged, which is insufficient for later preservation and transportation [12,15,16]. When the end-effector is in continuous contact with the fruit, mechanical bruises cannot be avoided regardless of the picking patterns. The elastic material used to contact the fruit which has been proofed to reduce bruising, and this method could be an easier method to implement during robotic harvesting of cherry tomatoes using a pulling pattern [17–20]. Detachment force of fruit is greatly affected by the location of the separation point, stem–branch characteristics, and picking patterns [21,22]. The detachment force for sweet peppers [23], pumpkins [24], citrus [25], apples [26], and other fruits [15,27–29] was measured, and the relation between the detachment force and swing angles, gripping force, or bending directions was analyzed in different research.

It was observed from the above literature that the issue of fruit plucking patterns is of great significance for robot harvesting, according to different picking objects. With the fruit and stem as the research object, an enormous number of studies have been discussed on fruit plucking patterns (pulling, twisting, and bending) and different picking methods, including cutting, saw cutting, negative pressure adsorption mode, laser cutting mode, etc. [14,30,31]. In addition, the cherry tomatoes harvested using the above picking patterns usually contain calyxes and stems, and the fruits with calyxes and stems cannot be eaten directly by people after simple cleaning. Especially in the harvest season, a large number of harvested cherry tomatoes are processed into ketchup or canned by food product factories, but these factories do not want to spend more manpower, material resources, and financial resources to remove the calyxes and stems on the fruit. Therefore, it is necessary to successfully separate the fruit from the plant at the calyx/fruit joint for robotic cherry tomato harvesting. On the basis of the literature review, the following gaps were identified with respect to the pulling pattern of cherry tomato harvesting. (1) Researchers have investigated a limited number of cherry tomato harvesting parameters at the calyx/fruit joint in the pulling pattern. (2) The harvesting parameters of cherry tomato, harvested at the calyx/fruit joint with minimum separation force, have not been reported. (3) Predictive modeling for the effect of different grasping angles and harvesting angles on the detachment force of cherry tomato at the calyx/fruit joint has not yet been discussed.

Detachment force is one of the most important parameters for designing end-effectors. The smaller detachment force makes the end-effector design smaller, more flexible, and more compact, and better adapted to the complex picking environment, so as to improve the success rate of picking. It was found that the experimental optimization method could effectively reduce the detachment force of fruits. Most researchers have optimized single objectives or multi objectives on different picking methods by optimization techniques such as response surface methodology, or statistical designed experiments in optimizing robotic fruit harvesting [26,32]. Therefore, in the present work, response surface methodology (RSM) based on the Box–Behnken design (BBD) technique has been used to optimize the

parameters such as different harvesting angles and grasping angles so that minimum detachment force is obtained. RSM is widely used as one of the most useful statistical methods since it is targeted at the optimization of various parameters in the agricultural field, the manufacturing field, and in other fields [26,33,34]. In the present study, stability pulling harvesting pattern tests of three angle parameters (grasping angle, horizontal angle, and pitching angle) at three levels have been diverged to find their effect on the detachment force of cherry tomato harvesting at the calyx/fruit joint. An RSM-BBD-based quadratic model was developed to study the detachment force of cherry tomato harvesting. Optimization and desirability approaches have been used to obtain the optimal combination of harvesting angles and clamping angles. Finally, cherry tomatoes without residual calyxes and stems were harvested.

In this study, cherry tomatoes were taken as the research objects, and experimental and theoretical analyses of different grasping angles and harvesting angles are presented. In Section 2, we start with a description of the cherry tomato samples, main equipment, and picking pattern. Then, the experimental methods, experimental design, and results are provided in Section 3. Next, according to the experimental results, the effect of different grasping angles and harvesting angles on the detachment force of cherry tomatoes at the calyx/fruit joint are discussed in Section 4. Finally, the conclusions of the research are given in Section 5.

## 2. Materials and Methods

### *2.1. Cherry Tomato Samples*

Fresh "No.1 Hybrid" cherry tomatoes cultivated at a plant factory in Hefei (Anhui Province, China) were used for harvesting testing in this research. A total of 170 cherry tomato samples were randomly selected in December 2020 and experiments were conducted at the harvesting date. The structure of cherry tomato-stem system is shown in Figure 1a; basic shape parameters of the cherry tomato samples, including the fruit height (H), the equator diameter ($D_1$), the mass (M), the fruit stalk diameter ($D_2$), the length of calyx branch (L), and the ultimate breaking force of calyx branches (F) are described in Figure 1b and reported in Table 1, although researchers have identified these parameters as having minimal influence on the detachment force [35,36]. The experiment was carried out in a plant factory, and each cherry tomato picking direction was carried out by picking robots, as shown in Figure 2. Before the experiment, to ensure that the separation point was at the joint between the calyx and the fruit for cherry tomato harvesting, rather than at the separation layer, the upper parts of the pedicel and the fruit stalk were reinforced by super glue. During the pull test, the tension moving speed (3 mm·s$^{-1}$), tension moving distance (30 mm), and grasping force (10 N) were kept constant while other process parameters such as the angle of the grasping direction and the angles of the picking direction were varied, as shown in Table 2. The tension translational velocity was set to 3 mm·s$^{-1}$, so as to ensure that the fruit-stem-branch system was tested under a quasi-static load. A total of 17 pulling picking trial runs were performed. Thereafter, the separation force of cherry tomatoes between the calyx and the fruits was recorded by a force sensor. The average of ten values of detachment force at the same group was taken to effectively document the disparity in detachment force. All tests were performed within one day at a room temperature of 20 °C. All cherry tomato samples were measured with a vernier caliper with a resolution of 0.1 mm.

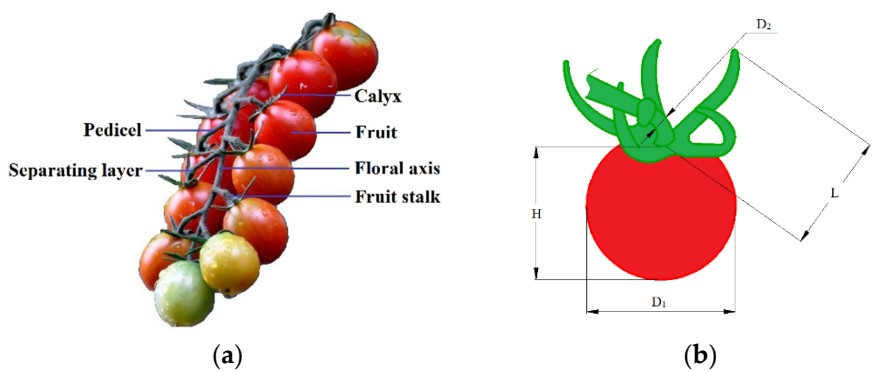

**Figure 1.** (**a**) The structure of a cherry tomato stem system; (**b**) physical measurements of cherry tomato basic shape parameters.

**Table 1.** Results of basic shape parameters of cherry tomato samples (values are means ± standard deviations).

| Parameter | H (mm) | $D_1$ (mm) | M (mm) | $D_2$ (mm) | L (mm) | F (N) |
|---|---|---|---|---|---|---|
| Value | 34.64 ± 5.63 | 29.77 ± 5.14 | 18.75 ± 6.52 | 2.10 ± 0.53 | 71.42 ± 3.96 | 0.57 ± 0.31 |

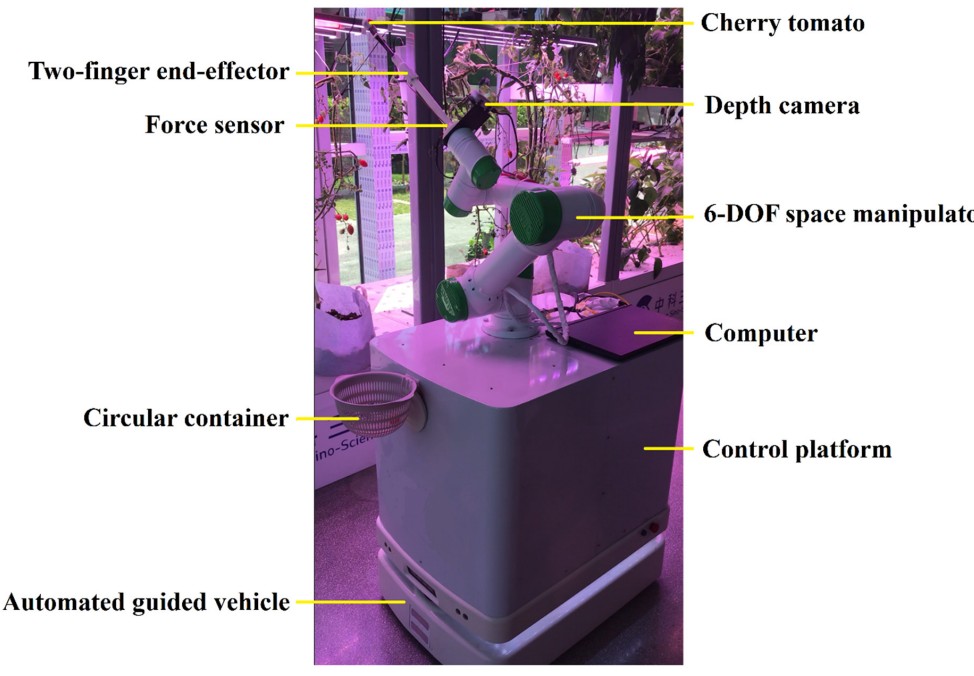

**Figure 2.** Harvesting robot for cherry tomatoes.

**Table 2.** Factors and levels of the Box–Behnken Design.

| Symbols | Input Variables | Units | Values | | |
|---|---|---|---|---|---|
| | | | Minimum | Medium | Maximum |
| $X_1$ | Grasping angle | ° | 30 | 60 | 90 |
| $X_2$ | Horizontal angle | ° | 45 | 90 | 135 |
| $X_3$ | Pitching angle | ° | 0 | 40 | 80 |

### 2.2. Main Instruments and Equipment

The cherry tomato harvesting experiment was carried out by a picking robot. As shown in Figure 2, the picking robot is mainly composed of a two-finger end-effector, a force sensor (HP-200, HANDPI, Zhejiang, China), a 6-DOF space manipulator (I5, AUBO,

Beijing, China), a control platform, and an automated guided vehicle (IR-R200, SUNSPEED, Shandong, China). The two-finger end-effector is located at the end of the 6-DOF space manipulator, the force sensor is installed between the two-finger end-effector and the 6-DOF space manipulator, the bottom of the 6-DOF space manipulator is installed at the upper end of the control platform, and the control platform is fixed on the automated guided vehicle. The force sensor was factory calibrated with a sensitivity of $1.5 \pm 10\%$ mV·V$^{-1}$, a rated load of 100 N, a comprehensive error value less than or equal to $\pm 0.1\%$, and a non-linearity value less than or equal to 0.1%. The 6-DOF space manipulator was factory calibrated with a maximum working radius of 886.5 mm, a repeated positioning accuracy of $\pm 0.02$ mm, a weight of 24 kg, and a repeated positioning accuracy of $\pm 0.02$ mm. The automated guided vehicle was factory calibrated with a maximum speed of 2 m·s$^{-1}$, a load of 200 kg, a weight of 80 kg, a positioning accuracy of $\pm 10$ mm, and a turning radius of 0 m. The two-finger end-effector installed at the end of the 6-DOF space manipulator is used to harvest the target fruit according to the specified picking pattern.

*2.3. Experimental Method of Cherry Tomato Harvesting*

A total of 17 picking experiments were performed for robotic cherry tomato harvesting. At first, the picking robot used a laser inertial system to build a map of the test site and located itself based on the location of the target plant. The three-dimensional coordinates of each cherry tomato fruit were calculated using the image recognition and positioning system from an image obtained by a binocular depth camera mounted at the end of the 6-DOF space manipulator. Then, the coordinate information of the ripe fruit was delivered to the upper machine of picking robot. The upper computer sent the coordinate information of the ripe fruit to the lower computer randomly, which sent the grasping instruction and the angle adjustment instruction to the 6-DOF space manipulator and the two-finger end-effector, respectively. Afterward, the 6-DOF space manipulator and the two-finger end-effector cooperated with each other to harvest the fruit in the pulling pattern. Clearly, the pulling pattern is when the target fruit is grabbed by the two-finger end-effector and the two-finger end-effector moves away from its current position in the direction opposite to the separation force between the fruit and the plant.

## 3. Experimental Design
*Response Surface Design*

RSM, which defines the correlation between responses and control factors, has been widely applied for agricultural optimization research [37–39]. BBD with a randomized experimental sequence was used as the response surface design. Assuming that the shape of the cherry tomato is oval, the Cartesian coordinate system O-XYZ is established, centering on the geometric center of the cherry tomato, as shown in Figure 3. Based on single-factor test results, in the cartesian coordinate system O-XYZ, the grasping angle between the clamping force direction of the two-finger end-effector and the Z-axis (Grasping angle, X1), the horizontal angle between the projection of the rotation axis of the two-finger end-effector in the XOY plane and the rotation axis of the two-finger end-effector (Horizontal angle, X2), and the pitching angle between the projection of the axis of rotation of the two-finger end-effector in the XOZ plane and the axis of rotation of the two-finger end-effector (Pitching angle, X3), were identified as key independent variables for picking patterns on fruit detachment, as given in Table 2. During the testing of the pulling pattern, an acute angle was observed for the default condition, while obtuse angles occurred if the angle between the projection of the rotation axis of the two-finger end-effector and the rotation axis of the two-finger end-effector was greater than 90° in the XOY plane. Note that it is specified that the clockwise is the positive direction for facilitate measurement, and the direction of the two-finger end-effector clamping force and the axis of rotation of the two-finger end-effector are collinear with the center of the mass of the cherry tomato.

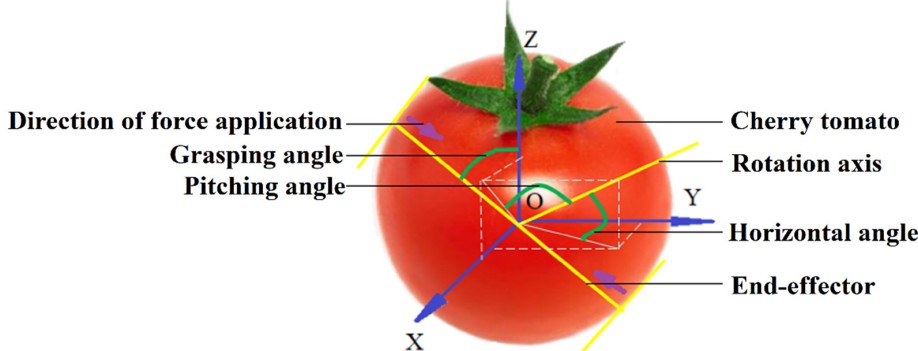

**Figure 3.** The relation of angle parameters in Cartesian coordinate system.

The experimental run was designed by Design Expert 6.0.8 software. Table 2 reports that the design matrix of three factors at three levels was obtained by applying BBD. According to the review of previous research, end-effector capability, and phenotypic omics of cherry tomatoes, the factors and levels were decided. The detachment force measured by the final conditions was considered as one of the response variables. A total of 17 testing groups were performed to model the empirical relationship among the key factor parameters in the pulling pattern by using RSM-BBD, and its mean is outlined in Table 3. The validity of the regression model was checked by ANOVA, which also determined the quadratic effect of the key factor parameters on the output response function.

**Table 3.** Experimental design made up of three independent variables at three levels, and the simulation results of the responses.

| Run | Input Parameters | | | Output Response |
|---|---|---|---|---|
| | Grasping Angle ($X_1$) | Horizontal Angle ($X_2$) | Pitching Angle ($X_3$) | Detachment Force (Y) |
| 1 | 90 | 90 | 80 | 2.21 |
| 2 | 60 | 135 | 0 | 0.58 |
| 3 | 60 | 90 | 40 | 1.41 |
| 4 | 60 | 45 | 0 | 0.67 |
| 5 | 60 | 135 | 80 | 1.81 |
| 6 | 30 | 135 | 40 | 1.88 |
| 7 | 30 | 90 | 0 | 1.18 |
| 8 | 60 | 90 | 40 | 1.79 |
| 9 | 60 | 90 | 40 | 1.86 |
| 10 | 60 | 90 | 40 | 1.73 |
| 11 | 90 | 90 | 0 | 0.66 |
| 12 | 90 | 135 | 40 | 1.88 |
| 13 | 60 | 90 | 40 | 1.55 |
| 14 | 90 | 45 | 40 | 1.84 |
| 15 | 30 | 90 | 80 | 2.46 |
| 16 | 30 | 45 | 40 | 1.91 |
| 17 | 60 | 45 | 80 | 1.92 |

## 4. Results and Discussions

Table 4 shows the analysis of variance for detachment force. The *p* value < 0.05 (at 95% acceptance level) suggest that the model is significant. The model *p* value 15.95 validates that the model was greatly significant. The *p* value of 0.80 shows that the lack of fit was insignificant. The chances of occurrence of "lack of Fit *p*-value" due to noise could be 67.48%. The Pred. $R^2$ of 0.6748 was in basically consistent with Adj. $R^2$ of 0.9535. Adeq. Precision measured the signal-to-noise ratio, and the ratio of 13.045 indicated a sufficient signal. Figure 4a shows that the effectiveness of developed model was confirmed by normal plots of residuals. It was also proven that the developed model has a high degree of accuracy in predicting the detachment force. The predicted value and the measured value were linearly fitted, and the fitting results were shown in Figure 4b. The final equation of the mathematical model of detachment force was given in Equation (1).

$$\text{Detachment force} = +1.95825 - 0.046383 \cdot X_1 + (6.49444 \times 10^{-3}) \cdot X_2 + 0.030294 \cdot X_3 + (1.29630 \times 10^{-5}) \cdot X_1 X_2 + (5.62500 \times 10^{-5}) \cdot X_1 X_3 - (2.77778 \times 10^{-6}) \cdot X_2 X_3 + (3.28889 \times 10^{-4}) \cdot X_1^2 - (4.27160 \times 10^{-5}) \cdot X_2^2 - (2.10312 \times 10^{-4}) \cdot X_3^2 \quad (1)$$

**Table 4.** The analysis of variance for detachment force.

| Items | Sum of Squares | DF | Mean Square | *f*-Value | *p*-Value |
|---|---|---|---|---|---|
| Model | 4.47 | 9 | 0.5 | 15.95 | <0.01 |
| X1 | 0.088 | 1 | 0.088 | 2.83 | 0.1364 |
| X2 | $4.51 \times 10^{-3}$ | 1 | $4.51 \times 10^{-3}$ | 0.14 | 0.7148 |
| X3 | 3.52 | 1 | 3.52 | 113.09 | <0.01 |
| X1X2 | $1.23 \times 10^{-3}$ | 1 | $1.23 \times 10^{-3}$ | 0.039 | 0.8485 |
| X1X3 | 0.018 | 1 | 0.018 | 0.58 | 0.4694 |
| X2X3 | $1.00 \times 10^{-4}$ | 1 | $1.00 \times 10^{-4}$ | $3.21 \times 10^{-3}$ | 0.9564 |
| X12 | 0.37 | 1 | 0.37 | 11.84 | 0.0108 |
| X22 | 0.032 | 1 | 0.032 | 1.01 | 0.3482 |
| X32 | 0.48 | 1 | 0.48 | 15.3 | <0.01 |
| Residual | 0.22 | 7 | 0.031 | | |
| Lack of Fit | 0.082 | 3 | 0.027 | 0.8 | 0.5533 |
| Pure Error | 0.14 | 4 | 0.034 | | |
| Cor Total | 4.69 | 16 | | | |
| Model Summary Statistics | | | | | |
| Std. Dev. | 0.18 | | R-Squared | 0.9535 | |
| Mean | 1.61 | | Adj R-Squared | 0.8937 | |
| C.V.% | 10.98 | | Pred R-Squared | 0.6748 | |
| PRESS | 1.53 | | Adeq Precision | 13.045 | |

Table 4 reports that the pitching angle was the most significant parameter affecting the detachment force, and it also contributes the most to the detachment force. These results are similarly reflected in the *p*-value of the pitching angle in Table 4. Figure 4c reports that the grasping angle gives a smaller detachment force of 1.66 N at a medium grasping angle. This was primarily because of the fact that a small grasping angle does not permit the gripper to grasp the calyx, resulting in a reduced gripping angle, which increases the probability of gripping the calyx and further increases the risk of separation force. Although the horizontal angle (Figure 4d) does not affect the detachment force alone, its interactions with other process parameters such as the grasping angle and the pitching angle commonly affect the detachment force, as shown in Figure 4f, h. Figure 4e reports that the pitching angle gives a smaller detachment force of 0.67 N at a low pitching angle. With an increase in the pitching angle from 0° to 80°. The detachment force goes on increasing up to a certain value of the pitching angle (80°) with a detachment force of 2.00 N.

Figure 4f reports that the detachment force was continuously decreased with increase in grasping angle. With an increase in grasping angle from 30° to 90°, the detachment force goes on decreasing up to a certain value of the grasping angle (68°) with a detachment force of 1.86 N; the detachment force again increases to 2.07 N with a further increase in grasping angle. Figure 4g reports that a combination of low pitching angle and high grasping angle gives a smaller detachment force of 0.66 N compared with 1.84 N at a high grasping angle and low horizontal angle combination. Compared with a high grasping angle and low pitching angle combination, the interaction between the horizontal angle and pitching angle (Figure 4h) reveals a smaller detachment force of 0.58 N was obtained at a high horizontal angle and low pitching angle.

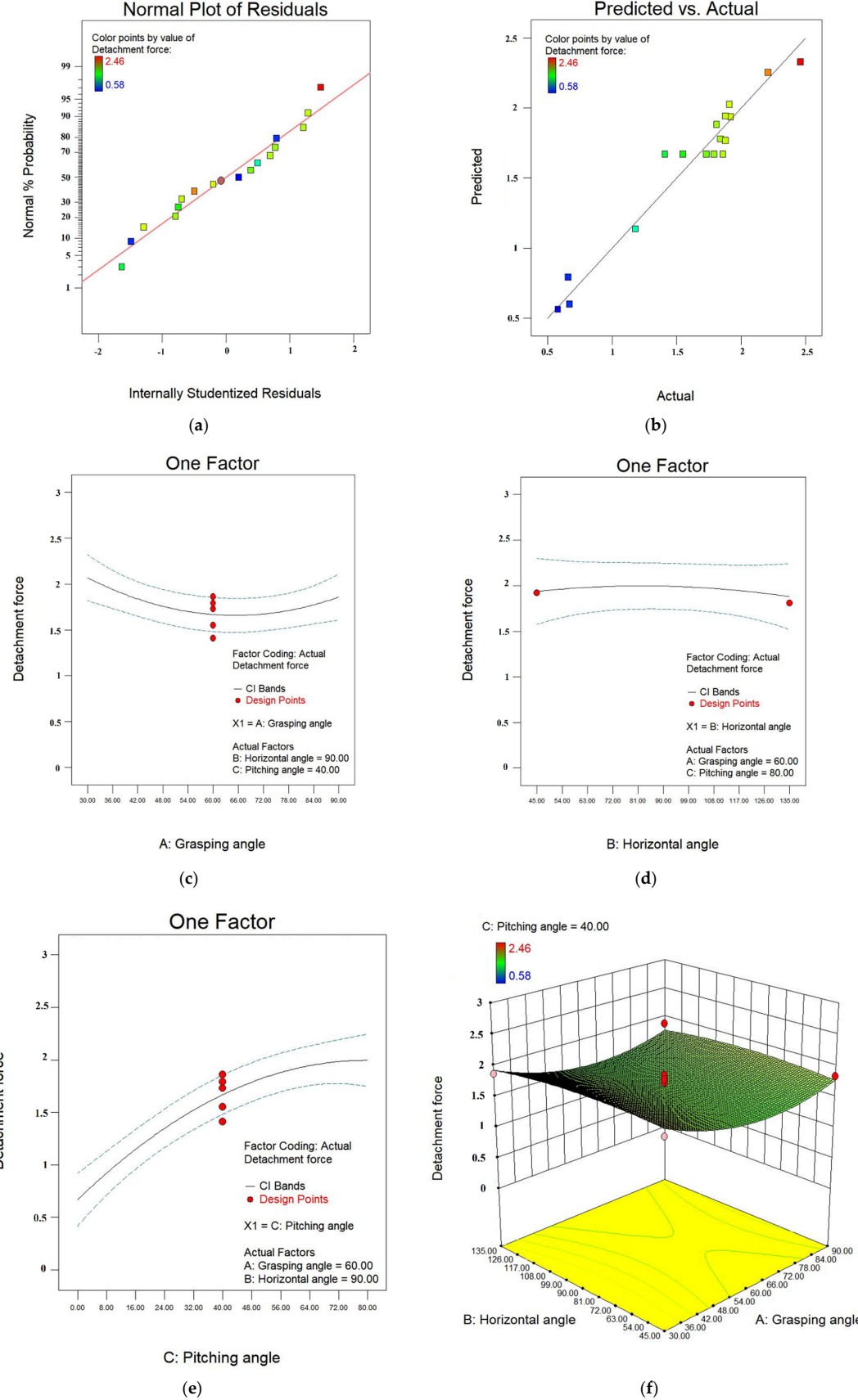

**Figure 4.** *Cont*.

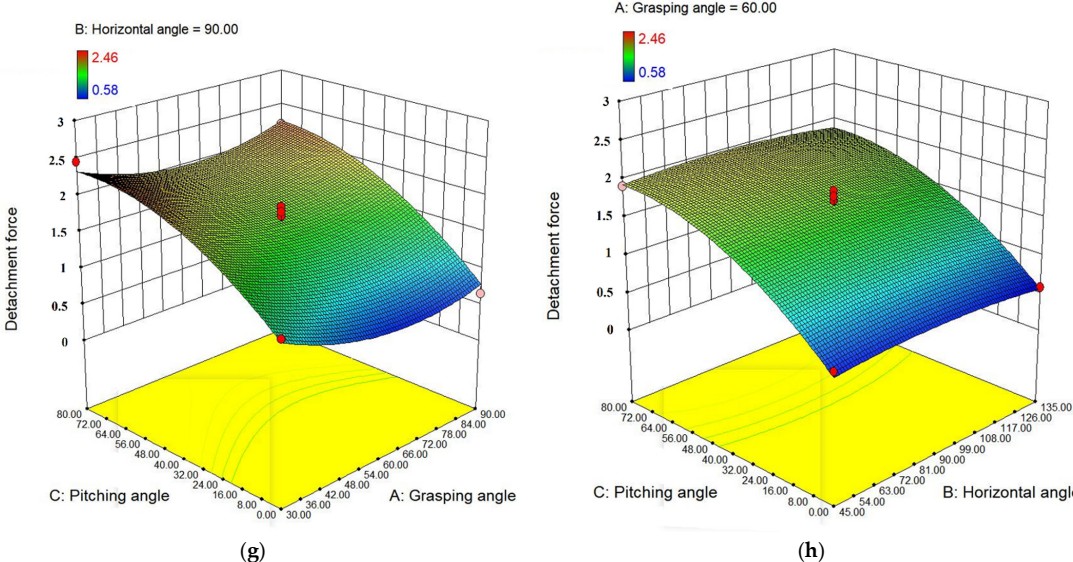

**Figure 4.** (**a**) Residuals plot; (**b**) relationship between predicted value and actual value; (**c**) the effect of grasping angle on detachment force; (**d**) the effect of horizontal angle on detachment force; (**e**) the effect of pitching angle on detachment force; (**f**) interaction plots for the interactive effects of grasping angle, and horizontal angle on detachment force; (**g**) interaction plots for the interactive effects of grasping angle, and pitching angle on detachment force; (**h**) interaction plots for the interactive effects of pitching angle, and horizontal angle on detachment force.

In summary, the detachment force increased first with the increase in grasping angle, and then decreased increasingly after reaching a maximum value. The detachment force decreased and then increased with the increase in horizontal angle. Conversely, the pitching angle had the opposite effect, i.e., the detachment force increased and then decreased with the increase in pitching angle. The curvatures of the contour lines (Figure 4g,h) revealed the following order: Figure 4g > Figure 4f > Figure 4h, which shows that the interactive effect of the grasping angle and pitching angle on detachment force is of the highest significance, followed by the grasping angle and horizontal angle, and finally the horizontal angle and pitching angle. Significantly, this conclusion is fantastically consistent with the results (the *p*-values of horizontal angle and pitching angle, grasping angle, and horizontal angle, and horizontal angle, and pitching angle were 0.4694, 0.8485, and 0.9564, respectively), which were disclosed in Table 4.

The optimum conditions for obtaining the minimum detachment force were a grasping angle of 67.84°, a horizontal angle of 135°, and a pitching angle of 0°. However, considering the simplicity of the practical operation, a verification experiment was carried out by adjusting the parameters: the grasping angle at 68°, the horizontal angle at 135°, and the pitching angle at 0°. The experimental detachment force (0.54 N) was greatly agreement with the predicted value (0.55 N).

The "smaller the better" desirability function was carried out to complete desirability analysis. The targets and ranges of input parameters grasping angle, horizontal angle, pitching angle, and the response viz. detachment force were given in Table 5. The main goal of the present study was to maximize the expected function by finding the optimal setting. Table 6 reports that the 25 optional desirability outcomes for detachment force were forecasted by the RSM-BBD-based model. Figure 5 reports the three-dimensional surface desirability plots. The experimental results show that detachment force varies from 0.58 N to 2.46 N. Moreover, five optimal solutions were randomly selected to verify the validity of the response surface equation. Each optimal solution was tested 10 times, and the mean value, standard deviation, and prediction error of the detachment force were calculated. The prediction RSM model and the results of the experimental test data were reported in Table 7, and the extremely low percentage of errors proves that the developed model was quite acceptable.

**Table 5.** Constraints of machine parameters and responses.

| Name | Goal | Lower Limit | Upper Limit | Importance |
|---|---|---|---|---|
| Grasping angle (°) | is in range | 30 | 90 | 3 |
| Pitching angle (°) | is in range | 45 | 135 | 3 |
| Horizontal angle (°) | is in range | 0 | 80 | 3 |
| Detachment force(N) | Minimize | 0 | 2.46 | 3 |

**Table 6.** Optimal combination for detachment force for higher desirability.

| Number | $X_1$ | $X_2$ | $X_3$ | Detachment Force (N) | Desirability |
|---|---|---|---|---|---|
| 1 | 67.84 | 135 | 0 | 0.54221 | 0.78 |
| 2 | 67.52 | 135 | 0 | 0.542246 | 0.78 |
| 3 | 67.27 | 135 | 0 | 0.542325 | 0.78 |
| 4 | 67.59 | 134.59 | 0.03 | 0.544893 | 0.778 |
| 5 | 65 | 135 | 0 | 0.544909 | 0.778 |
| 6 | 67.06 | 132.23 | 0 | 0.553648 | 0.775 |
| 7 | 73.8 | 135 | 0 | 0.553847 | 0.775 |
| 8 | 71.85 | 129.34 | 0 | 0.569333 | 0.769 |
| 9 | 69.56 | 45 | 0 | 0.569513 | 0.768 |
| 10 | 69.33 | 45.08 | 0 | 0.56983 | 0.768 |
| 11 | 71.39 | 45 | 0 | 0.570532 | 0.768 |
| 12 | 67.84 | 45 | 0 | 0.57057 | 0.768 |
| 13 | 68.75 | 45.39 | 0 | 0.571145 | 0.768 |
| 14 | 67.28 | 45 | 0 | 0.571327 | 0.768 |
| 15 | 65.97 | 45 | 0 | 0.573911 | 0.767 |
| 16 | 70.14 | 45 | 0.15 | 0.574725 | 0.766 |
| 17 | 74.14 | 45 | 0 | 0.576203 | 0.766 |
| 18 | 75.21 | 45 | 0 | 0.579754 | 0.764 |
| 19 | 69.12 | 45 | 0.43 | 0.584194 | 0.763 |
| 20 | 66.71 | 48.76 | 0 | 0.584924 | 0.762 |
| 21 | 56.45 | 135 | 0 | 0.584974 | 0.762 |
| 22 | 56.19 | 134.99 | 0 | 0.587001 | 0.761 |
| 23 | 69.68 | 50.4 | 0 | 0.587454 | 0.761 |
| 24 | 61.9 | 45.01 | 0 | 0.589164 | 0.761 |
| 25 | 69.46 | 56.03 | 0 | 0.603476 | 0.755 |

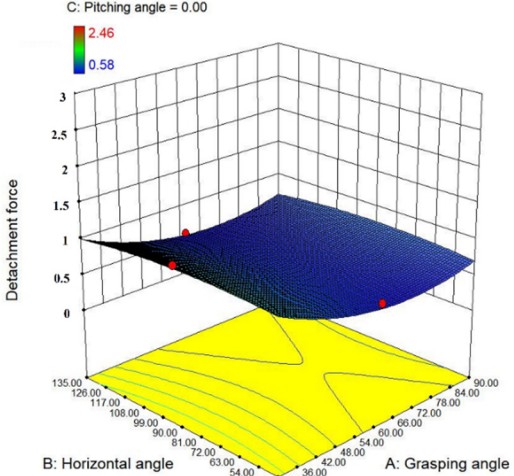

**Figure 5.** 3D desirability plot.

**Table 7.** Validation test results, including the mean, the standard deviation, and the prediction error of the experimental results.

| Exp. no. | RSM Model Prediction(N) | Experimental Results (N) | Prediction Error (%) |
|---|---|---|---|
| 1 | 0.542249 | 0.56 ± 0.02 | 2.5 |
| 2 | 0.544909 | 0.52 ± 0.03 | 1.9 |
| 3 | 0.573911 | 0.61 ± 0.05 | 3.2 |
| 4 | 0.593594 | 0.56 ± 0.03 | 4.2 |
| 5 | 0.600328 | 0.63 ± 0.05 | 3.7 |

RSM, response surface methodology.

## 5. Conclusions

The present study described the different angle parametric effect on the detachment force of cherry tomato harvesting at the calyx/fruit joint in a pulling pattern. Based on the grasping angle, horizontal angle and pitching angle, the quadratic models for detachment force have been marked out. A RSM-BBD experimental design was applied in this study to determine the optimal conditions for a minimum detachment force produced by the separation of a fruit from the plant at the calyx/fruit joint in a pulling pattern. The experimental design studies indicated that the pitching angle is the most important parameter, and the grasping angle has little effect on the detachment force. Moreover, the horizontal angle is not significant among the three parameters, but it has a significant interaction on the detachment force. The optimal combination conditions for obtaining a smaller detachment force were when the grasping angle was 68°, the horizontal angle was 135°, and the pitching angle was 0°, and the detachment force was found within the range of 0.58 N to 2.46 N.

In conclusion, this study is valuable for determining the influence of grasping angle, horizontal angle, and pitching angle on the smaller detachment force of cherry tomato harvesting at the calyx/fruit joint in a pulling pattern for cherry tomato harvesting, and providing an explanation of the interactions between these parameters. It is important to point out that this study can provide a reference for the picking of many fruits with a calyx/fruit joint, such as strawberries. Accordingly, this research provides significant perceptions and worthy theoretical methods for addressing this issue. In addition to the autonomous detection and the improvement of agronomy, to integrate robotic plucking with the genetic modification of cherry tomatoes is considered to be significant direction for future development.

**Author Contributions:** Conceptualization, H.X.; methodology, H.X. and D.K.; software, H.X. and J.S.; validation, H.X., D.K., J.S. and F.X.; formal analysis, H.X. and D.K.; investigation, H.X.; resources, D.K.; data curation, H.X.; writing—original draft preparation, H.X.; writing—review and editing, H.X. and D.K.; visualization, H.X.; supervision, D.K.; project administration, D.K. All authors have read and agreed to the published version of the manuscript.

**Funding:** This work was supported by the National Natural Science Foundation of China (Project No. 11774355) and the Fundamental Research Funds for the Central Universities (No. BC210202084).

**Institutional Review Board Statement:** Not applicable.

**Informed Consent Statement:** Not applicable.

**Data Availability Statement:** Not applicable.

**Acknowledgments:** The authors would like to express their sincere gratitude to Innovation Academy for Seed Design, Chinese Academy of Sciences, for their hydroponic cherry tomato samples.

**Conflicts of Interest:** The authors declare no conflict of interest.

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
