# Peer review of "Study the Parametric Effect of Pulling Pattern on Cherry Tomato Harvesting Using RSM-BBD Techniques"

_agriculture, doi:10.3390/agriculture11090815_

Round 1
Reviewer 1 Report
- Only 17 combinations of the levels of the factors (X1, X2, X3) were included in the experiment. Explain why the other combinations are omitted. Is this due to technical limitations or other reasons?
- The horizontal angle, as well as its interaction with the other two angles, are insignificant in the model (1). The authors' statement that "...its interaction with other process parameters such as the grasping angle, and the pitching angle significantly affect the detachment force as shown in Fig.4 (f, h)" does not seem to be sufficiently substantiated in the paper. "Significantly" means statistical significance. I can not see it. Perhaps my assessment is influenced by the lack of legibility of the Figures referred to by the authors.
- The quality of all Figures (except 1b) must be improved. The descriptions, as well as the values ​​on the axes are too small and therefore unreadable.
- In table 4, use the abbreviation d.f. or DF instead of DOF.
- Page 2 line 55: 23 s should be corrected to 23 [s / fruit]
- Page 2 line 64: "Mechanical bruises" or "mechanical bruises"?
- Revise the literature as required by the journal.
Author Response
Dear Reviewer:
Thank you for your letter and for the reviewers’ comments concerning our manuscript entitled “Study the parametric effect of pulling pattern on cherry tomato harvesting using RSM-BBD techniques” (ID: 1338419). Those comments are all valuable and very helpful for revising and improving our paper, as well as the important guiding significance to our researches. We have studied comments carefully and have made correction which we hope meet with approval. Revised portion are marked in red in the paper. The main corrections in the paper and the responds to the reviewer’s comments are as flowing:
Responds to the reviewer’s comments:
Reviewer #1:
Point 1: (Only 17 combinations of the levels of the factors (X1, X2, X3) were included in the experiment. Explain why the other combinations are omitted. Is this due to technical limitations or other reasons?)
Response 1: Firstly, Response surface methodology-box Behnken design (RSM-BBD) is an experimental condition optimization method, which is suitable for solving nonlinear data processing related problems. The method includes test design, modeling, testing the suitability of models and seeking the best combination of conditions (See 37-39 in the article for references). Secondly, RSM-BBD determines the approximate model through finite tests and response surface analysis to guide the subsequent design or test. Thirdly, for the three-factor at three levels test design, RSM-BBD was applied to a total of 17 groups of test design combinations. If there are four factors at three levels, a total of 29 experimental design combinations are applied by RSM-BBD. Finally, the key factors (X1, X2, X3) were determined according to the previous single factor experiment and experience in this article, the most significant influencing factors and the relationship between the key factors were determined. According to the analysis results, the subsequent cherry tomato picking strategy design and experiment were carried out.
Point 2: (The horizontal angle, as well as its interaction with the other two angles, are insignificant in the model (1). The authors' statement that "...its interaction with other process parameters such as the grasping angle, and the pitching angle significantly affect the detachment force as shown in Fig.4 (f, h)" does not seem to be sufficiently substantiated in the paper. "Significantly" means statistical significance. I can not see it. Perhaps my assessment is influenced by the lack of legibility of the Figures referred to by the authors.)
Response 2: We are very sorry for the misunderstanding caused by our misnomer. Line 245-247, the statements of “...its interaction with other process parameters such as the grasping angle, and the pitching angle significantly affect the detachment force as shown in Fig.4 (f, h)” were corrected as “...its interaction with other process parameters such as the grasping angle, and the pitching angle commonly affect the detachment force as shown in Fig.4 (f, h)”
Point 3: (The quality of all Figures (except 1b) must be improved. The descriptions, as well as the values ​​on the axes are too small and therefore unreadable.)
Response 3: As Reviewer suggested that the quality of all Figures must be improved. We have improved the quality of all the Figures, and the description and the values on the axes are enlarged for easier reading.
Point 4: (In table 4, use the abbreviation d. f. or DF instead of DOF.)
Response 4: As the Reviewer’s suggestion, we have used the abbreviation DF instead of DOF.
Point 5: (Page 2 line 55: 23 s should be corrected to 23 [s / fruit])
Response 5: As the Reviewer’s suggestion, the statements of “23 s” were corrected as “23 s / fruit”.
Point 6: (Page 2 line 64: "Mechanical bruises" or "mechanical bruises"?)
Response 6: As the Reviewer’s suggestion, the statements of “Mechanical bruises” were corrected as “mechanical bruises”.
Point 7: (Revise the literature as required by the journal.)
Response 7: As the Reviewer’s suggestion, we have revised the literature as required by the journal.
We tried our best to improve the manuscript and made some changes in the manuscript. These changes will not influence the content and framework of the paper. We appreciate for Reviewers’ warm work earnestly, and hope that the correction will meet with approval. Once again, thank you very much for your comments and suggestions.

Reviewer 2 Report
In this study, the authors consideration the picking pattern and the location where the fruit is separated are the key factors for separating the fruit from the plants. Although, it was found that there were differences in the detachment force and efficiency of different plucking patterns, moreover the effect of picking angle on the harvesting separation power of cherry tomato at calyx-fruit joint in pulling pattern have not been in-depth carried out. Therefore, this paper presents an experimental investigation to ascertain the effect of different grasping or harvesting angles on the detachment force of cherry tomato at calyx-fruit joint. Experiments were designed according to response surface methodology-box Behnken design by maintaining three levels of three process parameters—grasping angle, horizontal angle and pitching angle. The separation force at calyx-fruit joint was measured during cherry tomato harvesting. The design expert software was used to establish an optimized mathematical model of angle parameters for achieving the required detachment force. The manuscript represent the opinion of authors who consider that the results revealed that the minimum separation force of the cherry tomato harvesting at calyx-fruit joint was obtained by the addition of the grasping angle 68°, the horizontal angle 135° and the pitching angle 0°.
This may be interesting, but some important points need to be resolved. Importantly, a study must provide a critical analysis of the data. In other words, you must assess whether specific data published really stand up to scientific scrutiny. In order to achieve the above, you must clearly define your specific aims and objectives. So in your study you must develop a critical appraisal of the state of the art. This is an essential element of any article. There are important scientific questions (both conceptual and methodological) which need to be addressed with the primary studies. A study must highlight this. The introduction, which is written in clear language, covers a large number of relevant issues. Information are noteworthy, and are correct supported by similar results from the specialty (see WOS:000601854300001; WOS:000454169000026; WOS:000463708100021). Try to rewrite the abstract and conclusions, I also recommend the nuance of the introduction, the way of working is not very well explained, the procedure is tedious and unsustainable. For this reason, I recommend that the authors try to use more sustainable methodologies, the interpretation of the results can be improved/ reformulated
Author Response
Dear Reviewer:
Thank you for your letter and for the reviewers’ comments concerning our manuscript entitled “Study the parametric effect of pulling pattern on cherry tomato harvesting using RSM-BBD techniques” (ID: 1338419). Those comments are all valuable and very helpful for revising and improving our paper, as well as the important guiding significance to our researches. We have studied comments carefully and have made correction which we hope meet with approval. Revised portion are marked in red in the paper. The main corrections in the paper and the responds to the reviewer’s comments are as flowing:
Responds to the reviewer’s comments:
Reviewer #2:
Point: (This may be interesting, but some important points need to be resolved. Importantly, a study must provide a critical analysis of the data. In other words, you must assess whether specific data published really stand up to scientific scrutiny. In order to achieve the above, you must clearly define your specific aims and objectives. So in your study you must develop a critical appraisal of the state of the art. This is an essential element of any article. There are important scientific questions (both conceptual and methodological) which need to be addressed with the primary studies. A study must highlight this. The introduction, which is written in clear language, covers a large number of relevant issues. Information are noteworthy, and are correct supported by similar results from the specialty (see WOS:000601854300001; WOS:000454169000026; WOS:000463708100021). Try to rewrite the abstract and conclusions, I also recommend the nuance of the introduction, the way of working is not very well explained, the procedure is tedious and unsustainable. For this reason, I recommend that the authors try to use more sustainable methodologies, the interpretation of the results can be improved/ reformulated)
Response: We have rewritten the abstract and conclusions according to your comments. In addition, we deleted the relevant passage since they are not essential to the contents of the paper. Some contents were added to the experimental results to improve the interpretation of the results. We tried our best to improve the manuscript and made some changes in the manuscript. These changes will not influence the content and framework of the paper. And here we did not list the changes but marked in red in revised paper. Finally, we appreciate for Reviewer’ warm work earnestly, and hope that the correction will meet with approval.
Once again, thank you very much for your comments and suggestions.
